

# One-year effect of wearing orthokeratology lenses on the visual quality of juvenile myopia: a retrospective study

Yewei Yin[1], Yang Zhao[1], Xiaoying Wu[1], Mengyang Jiang[1], Xiaobo Xia[1], Yao Chen[1], Weitao Song[1], Shengfa Hu[1], Xia Zhou[1], Kelly Young[2] and Dan Wen[1]

[1] Department of Ophthalmology, XiangYa Hospital, Central South University, Changsha, Hunan, China
[2] Department of Veterans Affairs, Miami, United States of America

## ABSTRACT

**Objective**. To study the one-year effect of wearing orthokeratology (OK) lenses on the visual quality of juvenile myopia.

**Methods**. The right eyes of 36 juvenile myopias were retrospectively studied in this work. $Q$-value, $e$-value, corneal curvature, strehl ratio (SR), modulation transfer function (MTF) and wavefront aberration (WA) were compared before and at 1, 3 and 12 months after wearing OK lenses. The SR, MTF and WA of cornea, internal optic and ocular were analyzed separately. The spherical and cylinder diopter, vision acuity, compensating factor (CF) and compensative rate (CF%) were compared before and at 12 months after wearing OK lenses.

**Results**. (1) The vision of LogMAR increased and the corneal curvature decreased significantly after wearing OK lenses. There was no significant difference for the $e$-value before and after wearing OK lenses. The $Q$-value increased at 1 month but decreased at 3 and 12 months remarkably. (2) The ocular and internal optic SR and MTF increased significantly at 1 month and then remained stable. The MTF in different spacial frequencies increased after wearing OK lenses. There was no significant difference for the corneal SR before and after wearing OK lenses, and the corneal MTF decreased significantly after wearing OK lenses. (3) For the ocular, the total higher order aberration (HOA), spherical, coma and trefoil aberrations increased, and the total aberration, total lower order aberration (LOA) and defocus aberration decreased obviously except astigmatism. The corneal aberrations increased significantly after wearing OK lenses except astigmatism. For the internal optic, the total aberration, total LOA and defocus aberration decreased, and the total HOA, coma and trefoil aberration increased significantly except the astigmatism and spherical aberrations. (4) The CF and CF% of total aberration, total LOA, total HOA and coma aberrations increased, and those of astigmatism and spherical decreased at 12 months.

**Conclusions**. Orthokeratology is effective in correcting the refractive error and improving the vision quality of juvenile myopia over the one-year follow-up period.

Corresponding author
Dan Wen, 912096313@qq.com

**How to cite this article** Yin Y, Zhao Y, Wu X, Jiang M, Xia X, Chen Y, Song W, Hu S, Zhou X, Young K, Wen D. 2019.
One-year effect of wearing orthokeratology lenses on the visual quality of juvenile myopia: a retrospective study. *PeerJ* 7:e6998
http://doi.org/10.7717/peerj.6998

## INTRODUCTION

Orthokeratology (OK) is a non-invasive technology for refractive correction (*Cho et al., 2008*) and has been widely applied to treat young myopic patients since the 1990s (*Dave & Ruston, 1998*). It has been generally recognized that wearing this rigid contact lens at night can effectively reduce myopic degree (*Lian et al., 2014*; *Queiros et al., 2011*). With a reverse geometry design, the rigid contact lens is able to flatten the center of the front corneal surface by the pressure induced from the tear film and lids (*Queiros et al., 2010*). Meanwhile, the lens plays an important role in controlling myopia progression with a mechanism of myopia shift (*Walline, Jones & Sinnott, 2009*; *Cho & Cheung, 2012*). Moreover, the orthokeratic effect is reversible, and the cornea can recover to its original shape after stopping wearing the lens for 1 to 6 months (*Kobayashi et al., 2008*; *Chen, Lam & Cho, 2010*).

In general, most OK users could obtain an effective refractive correction in the daytime. However, there might be some side effects such as the night vision loss, ghosting, halos and glares, particularly within the first month after wearing the lens (*Santolaria et al., 2013*). Academically, there is still no consensus that wearing OK lenses can help improve the visual quality (*Zhong et al., 2015a*; *Ma et al., 2012*). The divergence might be associated with the amount of corrected refractive error, lens decentration and different evaluation methods, etc (*Li et al., 2017*; *Kaido et al., 2011*). In this work, the iTrace aberrometer was used to evaluate the visual quality objectively, which includes a few parameters that are with good reproducibility, i.e., the asphericity of cornea ($Q$-value), eccentricity of cornea ($e$-value), strehl ratio (SR), modulation transfer function (MTF) and wavefront aberration (WA) (*Win-Hall & Glasser, 2009*; *Chen et al., 2009*). Moreover, it is available to record the visual parameters respectively according to cornea, internal optic and ocular, which makes it possible to quantificationally assess the therapeutic effect of orthokeratology.

The main purpose of this work is to analyze the one-year effect of wearing OK lenses on the visual quality of juvenile myopia, and explore the possible reasons for the limited alleviation of vision quality. Corresponding statistical results could provide objective evidence to the improvement of visual quality after wearing OK lenses, and make benefits to the fitting of OK lens directly.

## PATIENTS AND METHODS

### Study subjects and methods

36 myopic teenagers were enrolled, and the parameters of their right eyes were analyzed retrospectively. The inclusion criteria was as follows: the age was 6~18 years old, and the spherical diopter was −1.00~−7.00D, and the cylinder diopter was 0D~−2.00D, and the treatment duration was more than one year with regular follow-up. The patients with any ocular surface diseases, systemic diseases or a history of wearing any contact lens before the therapy were excluded, and the teenagers who suffered any vision-threatening side events during the therapy were also excluded. The enrolled patients wore the OK lens no fewer than 8 h per night and removed the lens before 7 am on the next day. The follow-up examinations were performed between 8 and 10 am at each visit. The therapeutic process

lasted for more than 1 year from July 1st, 2015 to June 31st, 2017 in the vision care clinic of Xiangya Hospital, Central South University, China. All the subjects underwent the examinations including slitlamp microscope, manifest refraction and iTrace aberration examination before and at 1, 3 and 12 months after wearing OK lenses.

The spherical and cylinder diopter were measured by manifest refraction. The vision acuity was obtained using the standard logarithmic visual acuity charts. The slitlamp microscopy was used to detect the cornea. $Q$-value, $e$-value, corneal curvature, SR, MTF and WA were measured by the iTrace aberration analyzer (TRACEY VFA2 iTrace) when the pupil size was 4 mm. All the measurements were repeated three times for each eye. The compensating factor (CF) and compensative rate (CF%) were obtained from the WAs. CF was calculated by 1 minus the absolute ratio of ocular WA to corneal WA, and CF% referred to the ratio of positive CFs to all CFs. The spherical and cylinder diopter, the visual acuity, the CF and CF% were compared before and at 12 months, and the $Q$-value, $e$-value, corneal curvature, SR, MTF and WA were analyzed before and at 1, 3 and 12 months. The subjects' personal information was anonymized before the assessment.

This research project was reviewed and approved by the Ethic Committee of Xiangya Hospital of Central South University (the IRB approval number is 201505044). The design and methods of this research were in accordance with the requirements of regulations and procedures regarding to the human subject protection laws such as GCP and ICH-GCP. This study adhered to the tenets of the Declaration of Helsinki. Before wearing OK lenses, informed consents were obtained from the subjects and their guardians who fully understood the protocol and potential risks of the OK treatment. Permissions were also acquired to take the medical records of each subject as experimental data for the research purpose.

### Orthokeratology lens and lens fitting

The reverse-geometry OK lens (Japan alpha-ortho K) was applied in this work. The lens was manufactured with Boston EM material, with the central thickness of 0.22 mm, refractive rate of 1.422 and gas permeability of $104 \times 10-11$ $(cm^2 \times mlO_2)/(s \times mL \times mmHg)$. According to the simulated keratometry, corneal curvature and desired refractive change, the subjects were fitted with the lens using the lens-fitting software supplied by the manufacturers. The subjects wore the selected lens for one-hour trial.

### Statistical analysis

The IBM SPSS23.0 statistical packages and Microsoft Excel were used for the data management and analysis. The paired sample $t$-test was used to analyze the visual acuity, diopter and CF when the data had a normal distribution, and the Wilcoxon signed rank test was considered when the data distribution was abnormal. The one-way repeated measurement analysis of variance was applied to analyze the differences of the corneal curvature, $Q$-value, $e$-value, SR, MTF and WA before and at 1, 3 and 12 months after wearing OK lenses when the data followed the law of normal distribution; if not, Friedman test was used. $p < 0.05$ was considered statistically significant for the visual acuity, diopter and CF, and the adjusted alpha i.e., alpha/no. of test, was applied for the analysis with repeated measurement.
**Table 1  Descriptive statistics (mean ± SE) of myopic teenagers.**

|  | Before OK | 12 months after OK | $p$ |
|---|---|---|---|
| Gender | 19 female, 17 male | / |  |
| Age | $10.8 \pm 2.2$ | / |  |
| Sphere diopter | $-3.403 \pm 1.580$ | $-0.410 \pm 0.330$ | <0.001 |
| Cylinder diopter | $-0.465 \pm 0.539^{*}$ | $-0.210 \pm 0.210$ | 0.008 |
| LogMAR | $-0.788 \pm 0.242$ | $-0.008 \pm 0.177$ | <0.001 |

Notes.

*Four eyes had astigmatism above $-1.00$D.

**Table 2  Comparison of corneal curvature, $Q$-value and $e$-value.**

|  | Before OK | 1 month | 3 months | 12 months | $p$ |
|---|---|---|---|---|---|
| Horizontal corneal curvature, D | $41.065 \pm 1.781$ | $40.273 \pm 1.420$ | $40.384 \pm 1.336$ | $40.258 \pm 1.357$ | <0.001 |
| Vertical corneal curvature, D | $43.354 \pm 2.075$ | $41.406 \pm 1.419$ | $41.219 \pm 1.942$ | $41.320 \pm 1.859$ | <0.001 |
| $Q$-value | $-0.203 \pm 0.186$ | $0.375 \pm 0.643$ | $0.224 \pm 0.523$ | $0.202 \pm 0.532$ | <0.001 |
| $e$-value | $0.482 \pm 0.129$ | $0.528 \pm 0.211^{*}$ | $0.480 \pm 0.180^{*}$ | $0.424 \pm 0.210^{*}$ | 0.397 |

Notes.

The corneal curvature decreased significantly after wearing OK lens. $Q$-value increased at 1 month and decreased dramatically after 3 months. There were no significant differences for $e$-value before and after wearing OK lens. Data is expressed as mean ± SD.

*Compared with that before OK, $p$ > adjusted alpha.

## RESULTS

### Descriptive statistics of the subjects

The descriptive statistics of the myopic teenagers before and after wearing OK lenses were summarized in Table 1. 19 females and 17 males aged of ($10.8 \pm 2.2$) years old were enrolled in the study. It was informed that the spherical diopter decreased dramatically from ($-3.403 \pm 1.580$) to ($-0.410 \pm 0.330$) at 12 months after wearing OK lenses; whereas the evolution of the cylinder diopter was not obvious, relatively speaking. Moreover, the vision of LogMAR in the daytime increased significantly at 12 months. Two subjects complained about ghosting in the initial period of wearing OK lenses due to severe lens decentration, and this symptom gradually vanished after refitting the lens. Three subjects had night vision impairment in the late period, and a new pair of OK lenses was replaced after one year.

### Comparison of corneal curvature, $Q$-value and $e$-value before and after wearing OK lenses

Since the improvement of vision acuity originates from the reshaping of cornea, the corneal topography analysis is necessary for evaluating the effect of OK treatment. As shown in Table 2, under the pressure induced by the OK lens, both horizontal and vertical corneal curvatures reduced obviously after 1 month and kept stable subsequently. Meanwhile, $Q$-values increased at 1 month and gradually reduced at 3 and 12 months. For the $e$-value, no significant differences were noticed at 1, 3 and 12 months when compared respectively with that before wearing OK lenses.

**Table 3   Comparison of SR and MTF.**

|  | Before OK | 1 month | 3 months | 12 months | *p* |
|---|---|---|---|---|---|
| Ocular SR | 0.001 ± 0.001 | 0.017 ± 0.019 | 0.028 ± 0.033 | 0.017 ± 0.026 | <0.001 |
| Corneal SR | 0.051 ± 0.048 | 0.027 ± 0.020* | 0.028 ± 0.026* | 0.031 ± 0.023* | 0.237 |
| Internal optic SR | 0.001 ± 0.002 | 0.014 ± 0.012 | 0.020 ± 0.020 | 0.020 ± 0.026 | <0.001 |
| Ocular MTF, c/deg | 0.054 ± 0.018 | 0.141 ± 0.069 | 0.165 ± 0.083 | 0.138 ± 0.074 | <0.001 |
| Corneal MTF, c/deg | 0.265 ± 0.091 | 0.189 ± 0.068 | 0.184 ± 0.052 | 0.205 ± 0.072 | 0.001 |
| Internal optic MTF, c/deg | 0.062 ± 0.026 | 0.137 ± 0.056 | 0.165 ± 0.081 | 0.148 ± 0.089 | <0.001 |

**Notes.**

There were no significant differences of corneal SR before and after wearing OK lens, and the corneal MTF decreased after wearing OK lens significantly. After wearing OK lens the ocular and internal optic SR and MTF increased obviously. Data is expressed as mean ± SD.

*Compared with that before OK, *p* > adjusted alpha.

## Comparison of SR and MTF before and after wearing OK lenses

The objective evaluation of visual quality is necessary for the correction of refractive errors, and SR and MTF quantitatively represent the variation of vision quality. As summarized in Table 3, both the ocular and internal optic SR and MTF increased significantly at 1 month and kept stable at 3 and 12 months. It is noteworthy that the variation of corneal SR or MTF was different from that of ocular and internal optic. No significant differences existed for corneal SR before and after wearing OK lenses, and the corneal MTF decreased significantly after wearing OK lenses.

The assessment of MTF at different spatial frequencies is important for estimating the visual quality at different contrast levels. As illustrated in Figs. 1, 2 and 3, the evolution of ocular, corneal and internal MTFs were addressed respectively. The ocular and internal optic MTFs after wearing OK lenses were higher than those before wearing OK lenses at the same spatial frequency. Moreover, the ocular, corneal and internal optic MTFs after wearing OK lenses all decreased with the increase of the spatial frequency, which was similar with the evolution tendency of MTFs before wearing OK lenses.

## Comparison of wavefront aberrations before and after wearing OK lenses

### Comparison of ocular aberrations

The statistical results of ocular aberrations were listed in Table 4. One can see that the total higher order aberration (HOA), spherical, coma and trefoil aberrations increased obviously after wearing OK lenses, except the trefoil aberration at 12 months which had no statistical difference when compared with the baseline value. The total aberration, total lower order aberration (LOA) and defocus aberrations decreased dramatically after wearing OK lenses. One exception was the astigmatism aberration that had no obvious difference after wearing OK lenses.

### Comparison of corneal aberrations

As summarized in Table 5, most of the corneal aberrations increased significantly after wearing OK lenses. The exception was the post-treatment astigmatisms and the trefoil aberration at 12 months, which revealed limited difference when compared with the baseline value.

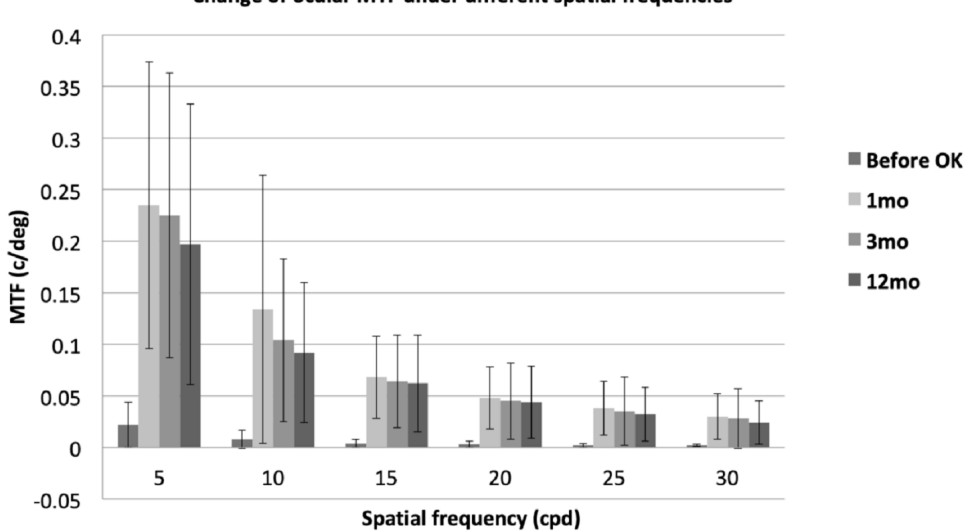

**Figure 1** The evolution of ocular MTF at different spatial frequencies before and at 1, 3, 12 months after wearing the OK lens. Error bar: standard devision. Change of ocular MTF under different spatial frequencies.

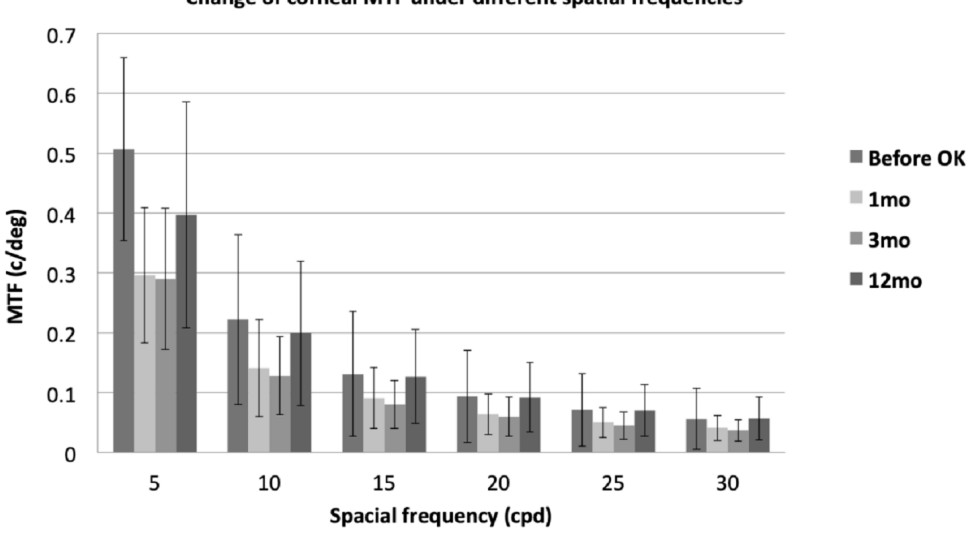

**Figure 2** The evolution of corneal MTF at different spatial frequencies before and at 1, 3, 12 months after wearing OK lens. Error bar: standard devision. Change of corneal MTF under different spatial frequencies.

### Comparison of internal optic aberrations

As shown in Table 6 that the total aberration, total LOA and defocus aberrations of internal optic decreased significantly, and the total HOA, coma and trefoil aberrations increased obviously after wearing OK lenses except the trefoil aberration at 12 months. It's remarkable
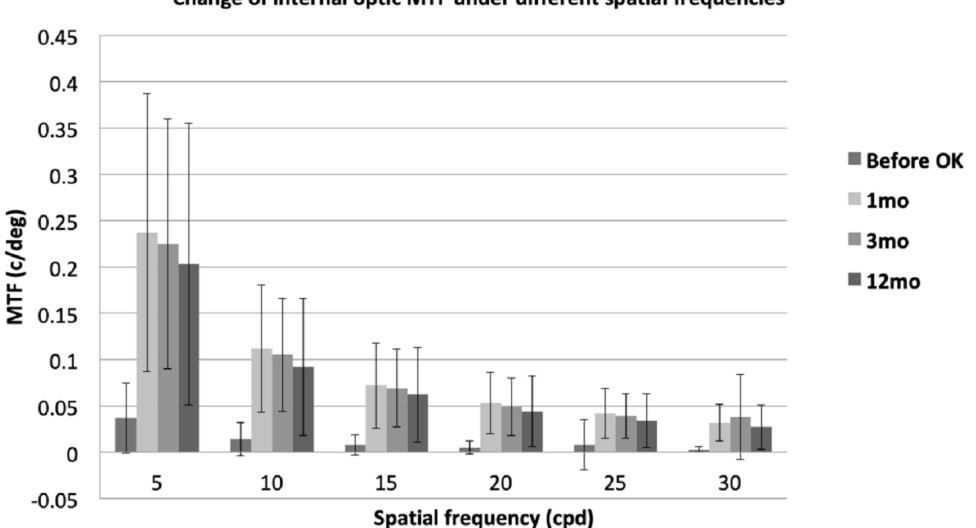

**Figure 3** **The evolution of internal optic MTF at different spatial frequencies before and at 1, 3, 12 months after wearing OK lens. Error bar: standard devision.** Change of internal optic MTF under different spatial frequencies.

**Table 4** **Comparison of ocular aberrations.**

|  | Before OK | 1 month | 3 months | 12 months | $p$ |
|---|---|---|---|---|---|
| Total aberration | 2.446 ± 1.055 | 1.518 ± 1.038 | 1.564 ± 1.164 | 1.741 ± 1.326 | <0.001 |
| Total LOA | 2.441 ± 1.054 | 1.371 ± 1.012 | 1.422 ± 1.126 | 1.628 ± 1.285 | <0.001 |
| Defocus | 2.404 ± 1.029 | 1.248 ± 1.051 | 1.328 ± 1.182 | 1.445 ± 1.252 | <0.001 |
| Astigmatism | 0.262 ± 0.119 | 0.392 ± 0.227[*] | 0.364 ± 0.215[*] | 0.372 ± 0.157[*] | 0.017 |
| Total HOA | 0.141 ± 0.068 | 0.496 ± 0.488 | 0.511 ± 0.328 | 0.497 ± 0.492 | <0.001 |
| Spherical | 0.032 ± 0.037 | 0.180 ± 0.191 | 0.189 ± 0.145 | 0.194 ± 0.182 | <0.001 |
| Coma | 0.075 ± 0.052 | 0.347 ± 0.253 | 0.358 ± 0.283 | 0.246 ± 0.144 | <0.001 |
| Trefoil | 0.049 ± 0.026 | 0.109 ± 0.092 | 0.137 ± 0.091 | 0.108 ± 0.084[*] | <0.001 |

**Notes.**
After wearing OK lens the total HOA, spherical, coma and trefoil aberrations increased, and the total aberration, total LOA and defocus aberration decreased. Data is expressed as mean ± SD.
*Compared to that before OK, $p$ > adjusted alpha.

that there were no significant differences for the astigmatism and spherical aberrations after wearing OK lenses when compared with the baseline value.

### Comparison of CF and CF% of aberrations

The CF of total aberration, total LOA, total HOA and coma aberrations increased significantly at 12 months, and the corresponding CF% also increased after wearing OK lenses. In contrast, there were no statistical significances for the CFs of defocus, astigmatism, spherical and trefoil aberrations when compared with that before wearing OK lenses (as summarized in Table 7).

**Table 5  Comparison of corneal aberrations.**

|  | Before OK | 1 month | 3 months | 12 months | $p$ |
|---|---|---|---|---|---|
| Total aberration | 0.544 ± 0.213 | 1.187 ± 0.980 | 1.199 ± 0.601 | 1.372 ± 2.038 | <0.001 |
| Total LOA | 0.529 ± 0.222 | 1.012 ± 0.754 | 1.050 ± 0.531 | 1.147 ± 1.650 | <0.001 |
| Defocus | 0.131 ± 0.125 | 0.670 ± 0.699 | 0.790 ± 0.600 | 0.865 ± 1.360 | <0.001 |
| Astigmatism | 0.472 ± 0.206 | 0.610 ± 0.361[*] | 0.492 ± 0.199[*] | 0.366 ± 0.214[*] | 0.040 |
| Total HOA | 0.108 ± 0.037 | 0.587 ± 0.657 | 0.544 ± 0.346 | 0.717 ± 1.220 | <0.001 |
| Spherical | 0.033 ± 0.026 | 0.151 ± 0.122 | 0.175 ± 0.114 | 0.211 ± 0.376 | <0.001 |
| Coma | 0.043 ± 0.031 | 0.435 ± 0.349 | 0.487 ± 0.323 | 0.414 ± 0.324 | <0.001 |
| Trefoil | 0.058 ± 0.029 | 0.116 ± 0.070 | 0.134 ± 0.076 | 0.111 ± 0.075[*] | <0.001 |

Notes.

The corneal aberrations increased significantly after wearing OK lens except astigmatism. Data is expressed as mean ± SD.

[*]Compared to that before OK, $p >$ adjusted alpha.

**Table 6  Comparison of internal optic aberrations.**

|  | Before OK | 1 month | 3 months | 12 months | $p$ |
|---|---|---|---|---|---|
| Total aberration | 2.287 ± 0.997 | 1.287 ± 0.854 | 1.201 ± 0.869 | 1.528 ± 1.457 | <0.001 |
| Total LOA | 2.309 ± 1.029 | 1.150 ± 0.874 | 1.087 ± 0.894 | 1.278 ± 1.132 | <0.001 |
| Defocus | 2.273 ± 1.035 | 0.584 ± 1.212 | 0.584 ± 1.185 | 0.570 ± 1.469 | <0.001 |
| Astigmatism | 0.320 ± 0.123 | 0.426 ± 0.253[*] | 0.436 ± 0.213[*] | 0.331 ± 0.174[*] | 0.008 |
| Total HOA | 0.154 ± 0.054 | 0.447 ± 0.321 | 0.413 ± 0.219 | 0.622 ± 1.080 | <0.001 |
| Spherical | −0.001 ± 0.037 | 0.030 ± 0.136 | 0.013 ± 0.092 | −0.017 ± 0.442 | 0.519 |
| Coma | 0.086 ± 0.049 | 0.286 ± 0.159 | 0.291 ± 0.193 | 0.319 ± 0.239 | <0.001 |
| Trefoil | 0.074 ± 0.026 | 0.166 ± 0.118 | 0.148 ± 0.092 | 0.112 ± 0.073[*] | <0.001 |

Notes.

The total aberration, total LOAs and defocus aberration decreased significantly after wearing OK lens, and the total HOAs, spherical, coma and trefoil aberration increased significantly. Data is expressed as mean ± SD.

[*]Compared to that before OK, $p >$ adjusted alpha.

**Table 7  Comparison of compensating factors and compensation rates for aberrations.**

|  | CF | | $p$ | CF% | |
|---|---|---|---|---|---|
|  | Before OK | 12 months |  | Before OK | 12 months |
| Total aberration | −4.013 ± 2.532 | −1.064 ± 1.788 | <0.001 | 0% | 36.1% |
| Total LOA | −4.407 ± 3.022 | −1.390 ± 2.155 | <0.001 | 0% | 30.6% |
| Defocus | −31.646 ± 54.817 | −32.438 ± 166.855 | 0.979 | 0% | 36.1% |
| Astigmatism | −0.329 ± 3.271 | −0.385 ± 0.970 | 0.921 | 86.1% | 38.9% |
| Total HOA | −0.379 ± 0.719 | 0.082 ± 0.435 | 0.001 | 33.3% | 61.1% |
| Spherical | −0.679 ± 3.793 | −1.621 ± 4.460 | 0.356 | 58.3% | 36.1% |
| Coma | −1.894 ± 2.538 | 0.226 ± 0.542 | <0.001 | 19.4% | 66.7% |
| Trefoil | −0.177 ± 0.974 | −0.142 ± 1.019 | 0.888 | 63.9% | 63.9% |

Notes.

The CF and CF% of total aberration, total LOA, total HOA and coma aberrations increased at 12 months. Data is expressed as mean ± SD.

## DISCUSSION

It is well known that orthokeratology is effective for the correction of refractive error (*Zhong et al., 2015b*; *Hiraoka et al., 2007*). In this work, the effectiveness was further verified by the increase of the vision of LogMAR, which relied on the reshaping of cornea that is represented by the variation of the corneal curvature as well as the $Q$-value and $e$-value. Moreover, the corneal curvature, $Q$-value and $e$-value remained unchanged for months after wearing OK lenses, which demonstrated the stable and long-term correction effect of OK.

However, the improvement or decline of visual quality after wearing OK lenses still remains in dispute. The eye is a complex and precise refractive system, and there are numbers of factors that affect the visual quality, e.g., the diffraction, scattering and refraction of light. Defined by the deviation between the actual and theoretical wavefront, wavefront aberration is known to reflect the influence of optical factors on the visual imaging. In this work, the objective assessment parameters of vision quality and wavefront aberrations were systematically measured and analyzed respectively based on 3 sections of cornea, ocular and internal optic. The purpose of the individual evaluation was to explore whether there were any differences and interactions between different parts of the optical system. Similar researches have been seldom reported in the published literature to the authors' knowledge.

Generally, the variations of ocular SR and MTF indicated that wearing OK lenses could help ameliorate the visual quality of the juvenile myopias, and the visual quality in different spacial frequencies was also improved. Individually, the decrease of corneal SR and MTF indicated the negative effect of OK associated with corneal reshaping. The small irregularity of cornea after reshaping can increase the light path of different parts of pupil, and thus produce higher corneal aberrations. And the aberrations of the ocular and internal optic changed differently from the corneal aberrations after wearing OK lenses. The various evolutions of wavefront aberrations demonstrated that the compensating effect worked in this phenomenon, which has been noticed in previous works (*Kelly, Mihashi & Howland, 2004*; *Artal et al., 2001*; *Gatinel et al., 2010*). The increased CF and CF% of some wavefront aberrations also supported this view. The high corneal aberration can be compensated by the intraocular effect, resulting in smaller aberration of the whole eye than that of the cornea. Moreover, the increase of internal optic SR and MTF might cover the decrease of corneal SR and MTF, which finally results in the increase of ocular SR and MTF and the improvement of visual quality.

However, the corneal astigmatism did not increase after wearing OK lenses. It was speculated that the stable corneal astigmatism was related to the uniformity of the remodeling force caused by the OK lens because the corneal curvature decreased synchronously along the vertical axis and horizontal axis. The compensating effect of internal optic astigmatism on the corneal astigmatism depended on the natural lens, and was related to the contraction of the ciliary muscles in all directions. As the corneal astigmatism did not increase, there was no need to activate the internal optic astigmatism to act as a neutralizing role. Therefore, both the ocular and internal optic astigmatism did

not increase. It also indicated that the compensation mechanism was a kind of dynamic compensation. The other two noticeable features were that the spherical aberration of internal optic remained unchanged after wearing OK lenses, and the trefoil aberrations of the ocular, internal optic and cornea at 12 months all decreased to the baseline value. It has been pointed out that the combination of trefoil, spherical aberration and vertical coma aberration might help promote the growth of the eyeballs (*Buehren et al., 2007*). Moreover, the vertical trefoil coupled with vertical coma could maximize the retinal image quality and reduce the deleterious impact of trefoil in visual performance (*Villegas, Alcon & Artal, 2007*). However, it is still not clear for the comprehension of the connection between the distinctive changes of spherical and trefoil aberration with the variation of eyeball and the visual quality, which deserves further exploration.

Furthermore, the compensation mechanisms are not clear so far (*Chen et al., 2017*; *Berrio, Tabernero & Artal, 2010*). One possible explanation might be the adaptive response of eye accommodation to corneal remolding, which primarily relies on the contribution of lens. The OK lens can reduce the hypermetropic refraction of the peripheral retina and improve the sensitivity and accuracy of accommodation that has a tight connection with the intraocular lens and ciliary muscle. With age, light, surgery or eye disease, this compensation mechanism may attenuate or even be broken, which might help comprehend the few inconformity of visual quality in some research.

Last but not the least, it should be noted that there still remain some limitations in the present work, e.g., some complications, including the lens decentration, dry eyes and corneal epithelium damage, were not well addressed. A potential selection bias was also introduced by the strict exclusion criteria about the adverse complications of wearing orthokeratology lenses. In addition, one can also try to increase the sample size and follow-up time to further verify the statistical results as observed in this work.

## CONCLUSION

In conclusion, orthokeratology is effective for the correction of refractive error and improvement of vision quality of juvenile myopia over the one-year follow-up period.

### Funding
This work was supported by the Natural Science Foundation of Hunan Province, China (No. 2015JJ4093) and the National Natural Science Foundation of China (No.81400400 and No.81770927). The funders had no role in study design, data collection and analysis, decision to publish, or preparation of the manuscript.

### Grant Disclosures
The following grant information was disclosed by the authors:
Natural Science Foundation of Hunan Province, China: 2015JJ4093.
National Natural Science Foundation of China: 81400400, 81770927.

## Competing Interests

The authors declare there are no competing interests.

## Author Contributions

- Yewei Yin conceived and designed the experiments, analyzed the data, prepared figures and/or tables, authored or reviewed drafts of the paper, approved the final draft.
- Yang Zhao, Mengyang Jiang and Yao Chen analyzed the data.
- Xiaoying Wu conceived and designed the experiments.
- Xiaobo Xia grammar checking.
- Weitao Song contributed reagents/materials/analysis tools.
- Shengfa Hu and Xia Zhou performed the experiments.
- Kelly Young contributed reagents/materials/analysis tools.
- Dan Wen conceived and designed the experiments, approved the final draft.

## Human Ethics

The following information was supplied relating to ethical approvals (i.e., approving body and any reference numbers):

This research project was reviewed and approved by the Ethic Committee of Xiangya Hospital of Central South University (201505044).

## Data Availability

Raw data is available as a Supplemental File.

## Supplemental Information

Supplemental information for this article can be found online at http://dx.doi.org/10.7717/peerj.6998#supplemental-information.

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
