# Peer review of "One-year effect of wearing orthokeratology lenses on the visual quality of juvenile myopia: a retrospective study"

_PeerJ, doi:10.7717/peerj.6998_

## Round 0.1 · original submission · Major Revisions

This is an interesting study, however, our reviewers have major concerns on the submitted version. Please kindly consider their comments and revise it accordingly.

Reviewer 1 ·

Basic reporting

The English expression need to be further polished, and abbreviations should be completely spelt out when they are first appeared, such as "HOA" and "LOA" in line 30.

Experimental design

1. The current study is a retrospective study and involved small numbers of people, which may make it less reliable.
2. According to the manuscript, during the one-year follow up, patients worn OK lens at night and removed them before 7 am on the next day (line 74). But authors didn't mention when patients worn OK lens (specific timepoint), so it's not sure if all patients worn OK lens in a fixed duration.
3. For optical quality measurement, pecific measure time wasn't mentioned in methods. Considering OK lens' influence to cornea would tail off with time, the measure time should be fixed to insure its veracity.

Validity of the findings

According to the result 3.4.1, astigmatism aberration had no statistical significance at 1 and 3 months, and was remarkablely higher at 12 months after wearing OK lens (line 149-151). But authors didn't explain this phenomenon or give possible mechanism in discussion, which may confuse the readers.

Reviewer 2 ·

Basic reporting

1. Please enhance the English writing of this study. There were many grammar errors.
2. Age of "10.806±2.184" years old was not an appropriate format for age.

Experimental design

1. Please describe the selection criteria in more details. What eye examinations were done?
2. The "parameters were analyzed before and at 1, 3 and 12 months". However, in Table 1, only data from baseline and 12 months were presented.
3. Please detail the statistical methods used in analyzing the repeated data. What type of student t-test was used? Since Friedman test was a kind of repeated measurement ANOVA, what another ANOVA was used? How was the repeated measurement ANOVA applied to the data? Did the authors checked the distribution of visual acuity and compensating factor?
4. How was the visual acuity measured?
5. As the orthokeratology lenses can pose potentially vision-threatening risk to the subjects, did the authors record any of the adverse events?
6. It is very important to know how the study subjects were selected.
7. Please define the error bars in the figures.
8. The authors should describe the compliance data during the study period. Any other treatments were used?
9. Did the authors obtain data regarding the known visual quality problems of orthokeratology lenses, such as night vision loss, ghosting, halos, and glares?
10. Correction for inflated alpha should be considered.

Validity of the findings

The authors over-interpreted the results. First, this study spanned for only 12 months. Second, most of the parameters were regressing to baseline values, which indicated that the treatment effect of the lenses was decreasing over time. Third, the authors did not report any side effects and subjective visual quality measurements. Fourth, the selection criteria and selection process were not clear.

Additional comments

This retrospective study provided potentially useful data for the field. However, there are some critical issues to be addressed.

·

Basic reporting

This is a well written, well executed, easy to read manuscript with no grammatical errors. The Figures and tables were self explanatory.

Experimental design

In this study the authors have made significant effort to study the visual quality of OK lenses worn juvenile patients who were followed up for a year.
The authors need to include the inclusion and exclusion criteria for recruiting the patients for this study.

Validity of the findings

As it has been mentioned in the discussion "the effect of OK lenses on control of myopia progression is well established", hitherto there is no or very little information is available on their effect on visual quality. This makes this study an unique piece of work, however the discussion part still needs to be re-written to highlight the robustness of this work and its significance to make it more interesting for the readers.

Additional comments

Once again I appreciate the authors for their effort taken to address visual quality of OK lenses.

---

## Round 0.2 · Minor Revisions

Some minor issues are still needed to be addressed. Please revise the paper accordingly. Before re-submission, please carefully check and edit the language of the paper all throughout again.

Reviewer 1 ·

Basic reporting

Professional English is used throughout. References, article structure and data mentioned in manuscript are accord with requirements.

Experimental design

No comment.

Validity of the findings

No comment.

Reviewer 2 ·

Basic reporting

Writing of the manuscript has been enhanced. Some important minor points should be address.

Experimental design

As the adverse events in wearing orthokeratology lenses are one of the major concerns, the authors should mention about excluding the cases who experienced any side effects of wearing the lenses. This information is important for the readers to know that the study population was selected partly based on clinical outcomes. Therefore, this should also be discussed as a limitation of this study. The authors also should be advised to acknowledge the potential biases introduced by the selection criteria.

Validity of the findings

The conclusion should be given a timespan of 12 months as shown in this population.

Additional comments

Thanks for the revision.

·

Basic reporting

The authors have made necessary changes in the manuscript taking Reviewer's comment into account.

Experimental design

I appreciate the author's for including the inclusion and exclusion criteria to make things clear.

Validity of the findings

The discussion part has been re-written in such a way to highlight the significance of the work with substantial citations to the previous work.

Additional comments

I appreciate the authors for making effort to address the Reviewer's comment.

---

## Round 0.3 · accepted · Accept

Thanks for your careful revisions.